# Profiles of stress hormones in relation to DENV serotypes among dengue-positive patients

Misbah Anwar[1][º], Abdul Waheed[1][º], Tehseen Fatima[2], Sadaf Khan[2], Mariyam Shahzad[2], Talat Roome[3], Ibrar Ahmed[4], Seil Kim[4], Zohaib Ul Hassan[4], Muhammad Aurongzeb[5], Mohammed Saeed Quraishy[6], Salman Ahmed Khan[1,2]*

1 Dow-DOGANA Advanced Molecular Genetics & Genomics Diseases Research & Treatment Centre, Dow University of Health Sciences, Karachi, Pakistan, 2 Department of Molecular Medicine, Dow College of Biotechnology, Dow University of Health Sciences, Karachi, Pakistan, 3 Dow Institute for Advanced Biological & Animal Research (DIABAR), Dow University of Health Sciences, Karachi, Pakistan, 4 Korea Research Institute of Standards and Science (KRISS), Daejeon, Republic of Korea, 5 Department of Biotechnology, Faculty of Engineering Science and Technology, Hamdard University, Karachi, Pakistan, 6 Dow University of Health Sciences, Karachi, Pakistan

º These authors contributed equally to this work.
* dr.salman@duhs.edu.pk

## Abstract

### Background

Dengue, a mosquito-borne illness caused by the Flavivirus and second only to COVID-19 in global impact, affects over 112 countries. With four distinct serotypes, current research emphasizes understanding genetic diversity and improving treatment and diagnostics. This study explores the relationship between stress hormones (cortisol and adrenaline) and DENV-1 sequence homology in dengue-positive patients.

### Methods

Among 165 subjects, hematological indices, NS1 antigen levels, and serotypes were analyzed. Hormone levels were quantified using ELISA, and viral load via RT-PCR. Amplicon sequencing and bioinformatics tools aided viral characterization. Statistical analyses included the Mann-Whitney U, Kruskal-Wallis, and Spearman's correlation tests for non-parametric comparisons.

### Results

The results revealed that cortisol and adrenaline levels were lower in dengue-positive patients, while patients infected with serotype DENV-2 showed comparatively higher epinephrine levels. DENV-1 was the most prevalent strain, accounting for 80% of cases. Hematological markers, including hemoglobin and red blood cell counts, showed comparable ranges. Phylogenetic reconstruction of DENV-1 whole-genome sequencing indicated multiple introductions into Pakistan from different countries, rather than a single source. The DENV-1 strain was found to predominantly affect

**Data availability statement:** All relevant data are within the paper and its Supporting information files.

**Funding:** The author(s) received no specific funding for this work.

**Competing interests:** The authors have declared that no competing interests exist.

males in Karachi. Notably, comparisons across different DENV serotypes did not reveal statistically significant differences in stress hormone response, indicating only descriptive trends. Spearman's correlation analysis showed weak, non-significant associations between viral load and cortisol (r = 0.184, p = 0.170) and between viral load and epinephrine (r = 0.165, p = 0.224). Statistically significant but negative correlation was observed only between viral load and hemoglobin (r = −0.228, p = 0.005).

## Conclusion

These findings enhance our understanding of the complex interactions between dengue serotypes and their impact on infection progression, particularly in relation to stress hormones and viral load. However, as most associations were exploratory and not statistically significant. The descriptive trends identified here highlight the need for larger, longitudinal studies to better define the relationship between stress hormones, viral load, and dengue severity.

## Background

Dengue is an acute viral illness caused by an RNA virus from the genus *Flavivirus* within the family *Flaviviridae.* The virus is arthropod-borne and transmitted primarily by *Aedes* mosquitoes. There are four antigenically distinct serotypes (DENV-1, DENV-2, DENV-3, and DENV-4) [1]. In 2007, a fifth serotype (DENV-5) was also identified in Sarawak, Malaysia [2]. Each serotype contains several subtypes or genotypes, based on mutations within the viral genome. While infection with one serotype provides lifelong immunity against that specific serotype, it only confers short-term, partial protection against others [3].

Dengue infection presents a wide spectrum of clinical symptoms, ranging from mild fever, headache, abdominal pain, and rash to more severe complications, such as dengue hemorrhagic fever (DHF) and dengue shock syndrome (DSS) [4]. Due to the growing global burden, the World Health Organization (WHO) regards dengue as a significant public health threat. Recent reports from WHO suggest that dengue is now the second most dangerous viral illness after COVID-19, owing to a sharp rise in infections in 2020 [2]. Dengue is endemic in 112 countries worldwide, with an estimated 100 million cases of dengue fever and 500,000 cases of DHF occurring annually. The case fatality rate for DHF ranges from 0.5% to 3.5%, particularly in Asia [5].

In Pakistan, dengue remains a persistent, year-round concern, especially in urban areas. Contributing factors include overcrowding, poor sanitation, and an influx of people into cities [6]. According to WHO reports, Pakistan witnessed 48,906 cases and 183 deaths due to dengue in 2021. Punjab recorded the highest number of cases (24,146), followed by Khyber Pakhtunkhwa (10,223). Sindh, Islamabad Capital Territory (ICT), Baluchistan, and Azad Jammu and Kashmir (AJK) also reported thousands of cases, with Karachi alone registering 1,255 cases [7].

Ongoing research is focused on various aspects of dengue, including vaccine development, diagnostic improvements, studying viral genetic variations, and exploring the impacts of climate change on transmission dynamics. Additionally, researchers are examining virus-host interactions and immune responses to better understand the pathophysiology of the disease. In this study, we aim to investigate the host immune response by assessing the levels of stress hormones-cortisol and adrenaline in dengue-infected individuals. In Indonesia, dengue control efforts are increasingly compromised by widespread resistance of Aedes aegypti to standard insecticides, notably pyrethroids and organophosphates, particularly across Java and neglected rural areas. This highlights the urgent need for sustainable alternatives such as plant-based biolarvicides [8].

The human endocrine system undergoes various physiological changes during viral infections. One notable change is the activation of the hypothalamic-pituitary-adrenal (HPA) axis by cytokines, leading to elevated levels of stress hormones such as cortisol, which influence the immune response. This interaction between viral infection, cytokine signaling, and HPA axis activation is well documented [9]. In dengue specifically, altered cortisol levels have been reported in relation to disease severity [10,11], although prior studies have not examined variation across serotypes or genotypes. Elevated pro-inflammatory cytokines such as TNF-α, IL-6, and IL-17, together with differential IL-10 responses in dengue hemorrhagic versus classical dengue, have been shown to play a major role in severity progression. These findings underscore the immune axis in dengue pathogenesis and provide theoretical support for exploring potential interactions between stress hormones and cytokine responses [12]. While adrenaline (or epinephrine) and norepinephrine facilitate the release of immune cells into the bloodstream, cortisol and adrenaline also play key roles in promoting immune cell differentiation and directing them to the tissues where they are needed [13]. The severe manifestations of dengue, such as sepsis and shock, are often accompanied by elevated cortisol levels. Although dengue has been widely studied in terms of epidemiology and viral diversity, little attention has been given to host endocrine responses. For example, cortisol has been linked to disease severity, with lower levels reported in children with severe dengue shock syndrome [10], but such studies did not assess differences by serotype or genotype. To date, no research has systematically evaluated stress hormone responses (cortisol and adrenaline) across DENV serotypes. Addressing this gap, our study investigates stress hormone profiles in relation to DENV serotype-genotype characterization in dengue-positive patients from Pakistan.

We have previously reported the whole-genome phylogenetic characterization of DENV-1 strains circulating in Pakistan [14]. Building on that foundation, the present study takes a more integrative approach. Specifically, we investigate host endocrine responses (cortisol and adrenaline) in dengue-positive patients and compare these with viral load, serotype distribution, and hematological indices.

Our objective is to determine the relationship between different dengue virus serotypes and stress hormone levels in the Pakistani population. By quantifying cortisol and epinephrine levels in dengue-positive patients and correlating them with dengue serotypes, we aim to enhance the understanding of dengue pathogenesis. This information could pave the way for treatments aimed at modulating the immune response to alleviate the impact of stress-induced immunological changes and aid in the prevention and control of dengue.

## Materials and methods

### Sample collection

This is a cross sectional observational study. A total of 165 participants, regardless of age or gender, were enrolled in this study. Fifteen healthy individuals provided control samples, while 150 suspected dengue patient samples were collected from Dow University Hospital, Karachi, between September 2022 and December 2022. During the outbreak, dengue-positive cases were recruited following clinical suspicion by physicians and confirmed through NS1 antigen and RT-PCR testing. Negative controls were recruited from individuals testing negative for dengue in the same setting, using random selection according to resource availability. Detailed fever history was not systematically available, which we recognize as a limitation. Although one-to-one matching was not possible, group-level comparability in age and sex distribution was

maintained, and baseline hematological indices were reviewed to minimize imbalances. Serum samples were stored at −80°C until further use. Official consent was obtained from the study participants and the study was approved from the ethical committee of Dow University of Health Sciences (IRB/DUHS/2024/40)

## NS1 antigen testing

Serum samples from suspected dengue patients were tested for the presence of dengue virus NS1 antigen using a one-step immunochromatographic assay with the Dengue NS1 rapid test cassette (CITEST Diagnostics Inc., Vancouver, BC, Canada), following the manufacturer's instructions. Positive samples, as indicated by the presence of test lines, were processed further for serotyping.

## Hematological parameters

Whole blood images of dengue-positive samples were analyzed using an automated cell analyzer (Beckman Coulter). Hematological parameters including hemoglobin levels, total leukocytes, red blood cells, and platelet counts were recorded for each patient.

## Serotyping

Viral RNA was extracted from 140 μL of blood using the QiAamp Viral RNA Mini Kit (Qiagen, Germany) according to the manufacturer's protocol. Extracted RNA was subjected to real-time RT-PCR using the Bosphore Dengue Virus Genotyping Kit v1. Reverse transcription was performed at 50°C for 30 minutes, followed by amplification. The amplification process was monitored using sequence-specific primers labeled with fluorescent molecules. Dengue serotypes 1 and 4 were detected using the FAM filter, while serotypes 2 and 3 were detected using the HEX filter. Internal controls were monitored using the CY5 channel to check for PCR inhibition and verify the isolation procedure. RT-PCR Cycle Threshold (Ct) values were recorded and used as a relative measure of viral load.

## ELISA for serum cortisol quantification

Serum cortisol levels were measured using ELISA kits (cat no. CEA462Ge.), following the competition-based assay protocol provided by the manufacturer. Briefly, 50 μL of sample and Reagent A were added to a microtiter plate and incubated at 37°C for 1 hour. After three washes with PBS, 100 μL of Reagent B was added, followed by incubation at 37°C for 30 minutes. After another washing step, 90 μL of substrate solution (TMB) was added to each well and incubated, followed by the addition of a stop solution (50 μL). The plate was immediately read at 450 nm, and the manufacturer's standard solution was used as a blank.

## Epinephrine (Adrenaline) determination

Epinephrine levels were determined using ELISA kits (cat no. CEA858Ge.), following a method similar to that used for cortisol quantification. Monoclonal antibodies specific to adrenaline were employed to quantify the hormone in both healthy individuals and dengue-infected patients.

## RNA extraction, cDNA synthesis, and sample shipment

Viral RNA was extracted from selected patient samples using the QIAamp Viral RNA Mini Kit (Qiagen, USA). RNA quantity and quality were assessed using a Nanodrop plate reader (SkanIt RE 4.1, Thermo Scientific, USA). cDNA synthesis was performed using the LunaScript® RT SuperMix (New England Biolabs) and RevertAid First Strand cDNA Synthesis Kit (Thermo Fisher, USA) as per the manufacturer's protocols. cDNA samples were shipped to Theragen (Korea) for total mRNA sequencing using the Illumina NovaSeq 6000 platform with a 150 PE sequencing run. Duplicate samples were

used for whole-genome amplicon sequencing at the Korea Research Institute for Standards and Science (KRISS), using the Oxford Nanopore GridION platform.

## Tiling PCR with DENV-1 primers panel

The study used a tiling PCR approach to amplify genomes for amplicon sequencing. Primer Panel 1 (odd-numbered primer pairs) and 2 (even-numbered primer pairs) were used, with eight cDNA samples quantified and adjusted to three concentrations 0.1 ng/ µL, 1 ng/ µL and 10 ng/ µL [9]. The PCR reactions were performed in triplicates, generating Nanopore sequencing data. Each Tiling PCR reaction involved 9 µL of total cDNA, followed by primer panels 1 and 2 (4.5 µL each) The Q5 Hot Start High-Fidelity 2X Master Mix was added to perform a two-step PCR under following conditions: initial denaturation at 98°C for 30 seconds (one cycle), followed by 35 cycles of denaturation at 95°C for 15 seconds, and annealing and extension at 63°C for 5 minutes. The reactions were pooled after quantification using the Quantus Fluorometer ONE dsDNA System (Promega, USA) before library preparation.

## Library preparation and nanopore sequencing

The PCR amplicons were purified using the QIAquick PCR Purification Kit (QIAGEN). PCR amplicons were processed using the NEBNext End Prep enzyme mix and NEBNext Ultra II End Repair reaction buffer to create blunt-ended DNA molecules by repairing damaged ends and adding d A-tails to the 3' ends. Barcodes were ligated using the Native Barcode Expansion Kit 1–96 (EXP-NBD196, Oxford Nanopore Technologies). Libraries were pooled and adapters ligated. After cleaning with AmpureXp Beads (Beckman Coulter, USA) the elute was prepared for sequencing by adding Sequencing Buffer (SQB) and Loading Beads (LB) from the SQK-LSK109 kit (Oxford Nanopore Technologies) using the Oxford Nanopore GridION platform, which performs real-time base-calling, generating DNA sequences in fastq format.

## Bioinformatics data analysis

The Oxford Nanopore data was analyzed using sequencing data from three triplicates of each sample. The quality of the data was assessed using FASTQC (version 0.12.0) [15], and initial 50 bases were trimmed using NanoFilt (v. 2.8.0) [16]. The reference genome of DENV-1 (NC_001477.1) was downloaded from NCBI and used for mapping using BWA (v. 0.7.17) [17] for both, Illumina and Nanopore data. Minimap2 (v. 2.17) [18] was also used for the Nanopore data to improve mapping efficiency. SAMtools (v. 1.10) [19] and BCFtools were used for consensus calling. seqtk (v. 1.3) was used to convert consensus sequence from fastq format into fasta format. The sequences were pairwise aligned with the reference DENV-1 genome using MAFFT in Geneious (v. 8.1.9) [20]. Annotations were transferred to the consensus sequences, and manual curation was done to remove ambiguous bases. BLAST search (NCBI) was performed to generate a multiple sequence alignment in using MAFFT, implemented in Geneious to reconstruct a maximum likelihood tree using W-IQ-TREE [21], available online at http://iqtree.cibiv.univie.ac.at/. The tree file in Newick format was used to reconstruct phylogeny using TreeDyn [22].

## Statistical analysis

Statistical analyses were conducted using SPSS version 22 (SPSS Inc., Chicago, IL, USA). Figures were generated using GraphPad Prism 8 (GraphPad Software Inc., La Jolla, CA, USA). Shapiro-Wilk test was used to analyze the data normality. Descriptive statistics were reported as median (range) for continuous variables and as numbers and percentages (%) for categorical variables. Mann-Whitney U test was used to compare differences between two continuous variables when the values were not normally distributed. To compare three groups that were categorically independent, Kruskal Wallis test was employed. In addition, when a statistical association was found, pairwise analysis was performed using Dunn's post hoc test with Bonferroni adjustment. Spearman's correlation was used to assess the associations between viral

load, stress markers, and hematological parameters in dengue-infected patients, and Bonferroni-adjusted p-values are reported. p-values less than 0.05 were regarded as statistically significant.

## Results

### Participants' demographic characteristics

Table 1 provides demographic data on the study participants. Fifteen healthy adults (11 females and 4 males) were included as controls. Among 150 NS1-positive dengue patients, 127 were male, and 22 were female. The patients had varying hematological profiles: 73 showed platelet counts below150 x1000 µL, while 77 had counts equal to or above this level. The range of WBC counts for these patients was 4–10 x1000/ µL, and only five patients exhibited hemoglobin levels below 12 g/dL.

### Dengue virus serotyping

Fig 2 Prevalence of DENV serotypes in Karachi, Pakistan.

Fig 1 DENV serotype distribution in male and female. The bar graph shows incidence of dengue virus strains in males and females from Karachi, Pakistan.

Following NS1-based dengue confirmation, all 150 samples underwent RT-PCR to determine the viral serotype. DENV-1 was most prevalent (80%), followed by DENV-2 (16%), with DENV-3 detected in only one case (0.7%). Co-infections of DENV-1 and DENV-2 were observed in 3.3% of patients (Table 2, Fig 1). Serotyping results showed a higher infection rate in males than females (Table 2, Fig 2).

**Table 1. Demographic Characteristics of Study Participants (n = 165).**

|  | Total Subjects n (%) | Healthy Individuals n (%) | Dengue-Infected Patients n (%) |
|---|---|---|---|
| **Number of Subjects** | 165 | 15 | 150 |
| **Age** |  |  |  |
| Children <18 years | 10 (6.1) | 0 (0.0) | 10 (6.7) |
| Adults >18 years | 155 (93.9) | 15 (100.0) | 140 (93.3) |
| **Gender (n = 164)** |  |  |  |
| Male | 131 (79.9) | 4 (26.7) | 127 (85.2) |
| Female | 33 (20.1) | 11 (73.3) | 22 (14.8) |
| **NS1 positive** | 150 | – | 150 |
| **Viral Load (Ct Value), Median (Range)** |  |  | 25 (18-34) |
| **Hematological Markers** |  |  |  |
| **Platelet Count (Ref Range: 150–410 x1000/µL)** | 150 | – | 150 |
| <150 x 1000/µL |  |  | 73 (48.7) |
| >150 x 1000/µL |  |  | 77 (51.3) |
| **Hemoglobin Levels (Ref Range: 12–15 g/dlL)** | 150 | – | 150 |
| <12 g/dL |  |  | 5 (3.3) |
| >12g/dL |  |  | 145 (96.7) |
| **Total Leucocyte Count (Ref Range: 4–10 x1000/µL)** | 150 | – | 150 |
| <4.0 x1000/µL |  |  | 47 (31.3) |
| >4.0 x1000/µL |  |  | 103 (68.7) |

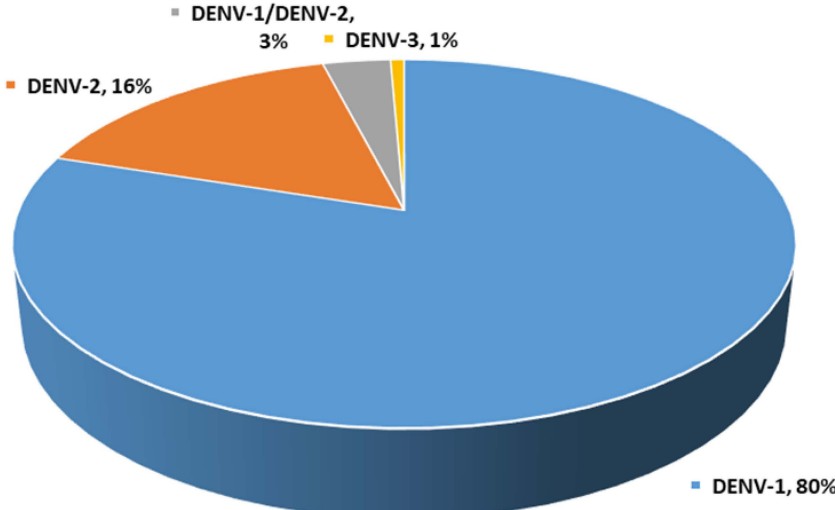

**Fig 1. Prevalence of dengue virus (DENV) serotypes in Karachi, Pakistan (n = 150).**

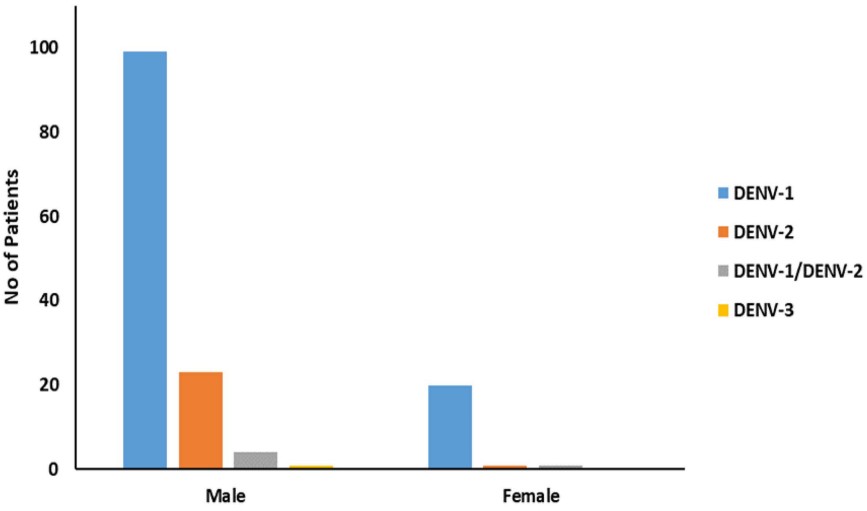

**Fig 2. DENV serotype distribution in male and female (n = 150).**

## Stress hormone levels

Table 3 shows the stress hormone levels, indicating significantly lower cortisol and epinephrine in dengue patients than healthy controls (p < 0.001). Median cortisol levels in healthy individuals were 177.0 ng/mL, while dengue patients showed a median of 79.7 ng/mL. Epinephrine levels were similarly reduced in infected individuals (90.6 pg/mL) compared to controls (537.4 pg/mL), with variations across serotypes as shown in Fig 3.

## Stress hormones and serotype-specific effects

The impact of each strain on stress hormone level is shown in Table 4. Those individuals who were co-infected with DENV-1/2 generate more cortisol 87.6 ng/mL than those who were infected with DENV-1 or DENV-2, producing 80.1 and

**Table 2. Serotype distribution of dengue virus (DENV) in samples collected from DENV infected patients (n = 150).**

| Serotype | Total Samples n (%) | Male n (%) | Female n (%) |
|---|---|---|---|
| **DENV-1** | 120 (80.0) | 99 (77.9) | 20 (90.9) |
| **DENV-2** | 24 (16.0) | 23 (18.1) | 1 (4.6) |
| **DENV-1/DENV-2** | 5 (3.3) | 4 (3.2) | 1 (4.5) |
| **DENV-3** | 1 (0.7) | 1 (0.8) | 0 (0.0) |

DENV-1 – Dengue virus 1 serotype, DENV-2 – Dengue virus 2 serotype, DENV-1/DENV-2 – Dengue virus 1/ Dengue virus 2 serotype, DENV-3 – Dengue virus 3 serotype.

**Table 3. Stress hormones (Cortisol and Epinephrine) levels in healthy and dengue-infected individuals (n = 72).**

| Stress Hormones | Healthy Individuals Median (Range) (n = 15) | Dengue Infected Patients Median (Range) (n = 57) | p-value* |
|---|---|---|---|
| **Cortisol Levels (ng/mL)** | 177.0 (104.0-227.5) | 79.7 (22.3-190.2) | <0.001 |
| **Epinephrine Levels (pg/mL)** | 537.4 (113.3-1077.4) | 90.6 (24.9-1430.1) | <0.001 |

\* p value <0.05 was considered significant.

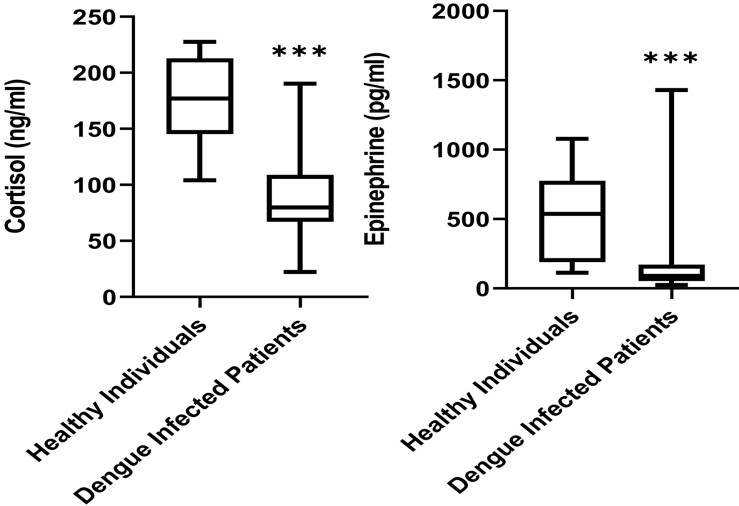

**Fig 3. Stress hormones levels in healthy individuals and dengue-infected patients (n = 72).**

76.5 ng/mL of cortisol, respectively. Infection with DENV-2 showed the highest levels of epinephrine, measuring 98.4 ng/mL, compared to infection with DENV-1(78.3 ng/mL) and simultaneous infection with DENV-1and 2 (51.5 ng/mL). In Fig 4, the box graphs for each hormone are displayed.

## Hematological parameters and serotype impact

Comparative hematological parameters across patients infected with dengue serotypes, as indicated in the Table 5, identified that patients who were infected with either DENV-1 or 2 or co-infected with both had nearly identical ranges of red blood cells and hemoglobin, which are 4.9 million/μl and 14.8 g/dl at maximum (p > 0.05). Patients with DENV-1 had

**Table 4. Stress hormones (Cortisol and Epinephrine) levels in healthy individuals and patients infected with different dengue virus (DENV) serotypes (n=71).**

| Study Groups | Cortisol Levels Median (Range) (ng/ml) | p-value*^ | Epinephrine Levels Median (Range) (pg/ml) | p-value*^ |
|---|---|---|---|---|
| Healthy Individuals (n=15) | 177.0 (104.0-227.5) | <0.001^ | 537.4 (113.3-1077.4) | <0.001^ |
| DENV-1 (n=32) | 80.1 (23.9-159.8) | | 78.3 (24.9-1430.1) | |
| DENV-2 (n=20) | 76.5 (22.3-190.2) | | 98.4 (58.6-775.1) | |
| DENV-1/DENV-2 (n=4) | 87.6 (69.7-131.9) | | 51.5 (48.6-180.3) | |
| **Comparison** | **Cortisol Levels p-value*#** | | **Epinephrine Levels p-value*#** | |
| DENV-1 vs Healthy | <0.001 | | <0.001 | |
| DENV-2 vs Healthy | <0.001 | | <0.013 | |
| DENV-1/DENV-2 vs Healthy | 0.154 | | <0.012 | |
| DENV-1 vs DENV-2 | 1.00 | | 1.00 | |
| DENV-1 vs DENV1/DENV-2 | 1.00 | | 1.00 | |
| DENV-2 vs DENV1/DENV-2 | 1.00 | | 1.00 | |

* p<0.05 is considered statistically significant.

^ Comparison between healthy individuals and different DENV serotypes, p-value computed using Kruskal–Wallis test.

# Dunn's pairwise comparison with Bonferroni adjustment.

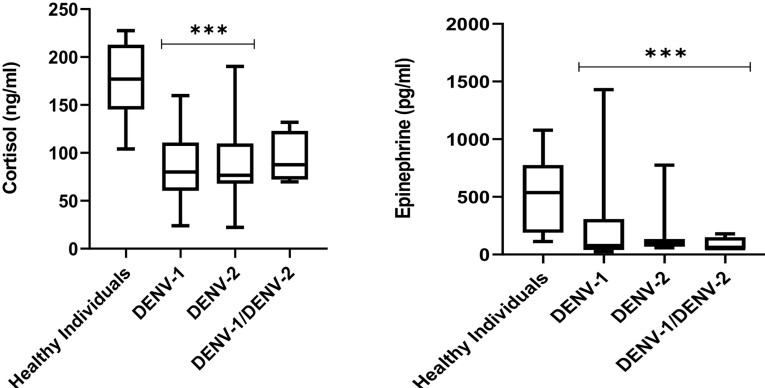

**Fig 4. Stress hormones levels in healthy individuals and patients infected with dengue virus serotypes (n=71).**

increased leukocyte production. The highest decrease in platelet count (141.0 x 1000/μl) was seen in DENV-2 infections (p>0.05).

## Correlation of viral load with stress hormones and hematological markers

The correlation was evaluated between the viral load and all stress hormones and hematological markers that were taken into account, as indicated in Table 6. A weak positive correlation was found between viral load and cortisol, epinephrine, leukocyte, and platelet counts, while hemoglobin and RBC counts correlated negatively. Only the associations between viral load, and hemoglobin was found statistically significant (p<0.05).

## Whole genome sequencing

**Quality and quantity of the RNA samples.** The RNA samples were assessed using a Nanodrop plate (Skanit RE 4.1, Thermoscientific), revealing high quality RNA with a 260/280 ratio of 1.88 to 2.2. The RNA quantity ranged from 104 to

**Table 5. Comparison of Hematological parameters of patients infected with different dengue virus (DENV) serotypes (n = 149).**

| Hematological Parameters | DENVV-1 Median (Range) (n = 120) | DENV2 Median (Range) (n = 24) | DENV-1/DENV-2 Median (Range) (n = 5) | p-value* |
|---|---|---|---|---|
| Hemoglobin (g/dL) | 14.4 (5.8-18.5) | 14.8 (10.6-16.7) | 14.5 (11.2-15.7) | 0.416 |
| Red Blood Cell (million/µl) | 4.8 (3.5-6.8) | 4.9 (3.6-6.3) | 5.0 (3.9-5.6) | 0.425 |
| Total Leukocyte Count (x 1000/µl) | 4.9 (1.7-16.9) | 4.1 (2.1-10.2) | 4.3 (2.6-8.4) | 0.815 |
| Platelets Count (x 1000/µl) | 152.0 (35-538) | 141.0 (65-282) | 170 (85-204 | 0.476 |

*p < 0.05 is considered statistically significant.

**Table 6. Correlation between Viral load, stress and hematological markers of dengue infected patients.**

| Stress and Hematological Markers | Viral Load (Ct Value) | | |
|---|---|---|---|
| | Correlation Coefficient (r) | p-value* | Bonferroni-adjusted p-value* |
| Cortisol Levels (ng/mL) | 0.184 | 0.170 | 1.000 |
| Epinephrine Levels (pg/mL) | 0.165 | 0.224 | 1.000 |
| Hemoglobin (Hb) (g/dL) | −0.228 | 0.005 | 0.030 |
| Red Blood Cell (million/µL) | −0.06 | 0.468 | 1.000 |
| Total Leukocyte Count (1000/µL) | 0.119 | 0.146 | 0.876 |
| Platelets Count (1000/µL) | 0.205 | 0.012 | 0.072 |

* p < 0.05 is considered statistically significant.

140 ng/µL. Despite the viral RNA extraction kit, contaminations were expected, therefore the sample quality was evaluated at various stages of the procedure for the subsequent phases of library creation and sequencing.

**Consensus calling using amplicon sequencing data from Gridion, Oxford nanopore platform.** The study recovered consensus sequences shorter than the original DENV-1 reference genome length at both ends, with 32 nucleotides from the 5' UTR and 116 nucleotides from the 3' UTR not included. As a result, the assembled genomes are draft due to trimming of poor-quality data. However, Trimming did not affect sequence recovery in genomes from coding regions due to overlapping sequences in primer pools 1 and 2, as shown in multiple sequence alignment (Fig 5).

## Phylogenetic analysis

Maximum likelihood phylogenetic analysis of DENV-1 sequences was performed using IQ-TREE2 (ver. 2.0.7) with the best-fit evolutionary model selected by ModelTest-NG, following the optimized pipeline previously reported in our earlier study [14]. The best-fit model for phylogenetic analyses of DENV-1 samples was TIM2 + F + R, with three distinct clusters calculated by the ModelFinder [23] tool implemented in W-IQ-TREE. Node confidence was assessed using 1,000 ultrafast bootstrap replicates, and the resulting tree (Supplementary, S1 Fig) suggested multiple introductions of DENV-1 into Pakistan. The samples from NIH, Islamabad, Pakistan, belong to distinct lineages, indicating multiple introductions from other countries rather than a single introduction in Pakistan.

## Discussion

Dengue virus (DENV) remains a fast-spreading global health threat, particularly in endemic regions like Pakistan, where cases have risen markedly over the past decade [24]. Clinical severity can range from mild fever to severe manifestations involving hemorrhage and organ damage, or plasma leakage. If treatment is delayed, there will be a high risk of morbidity

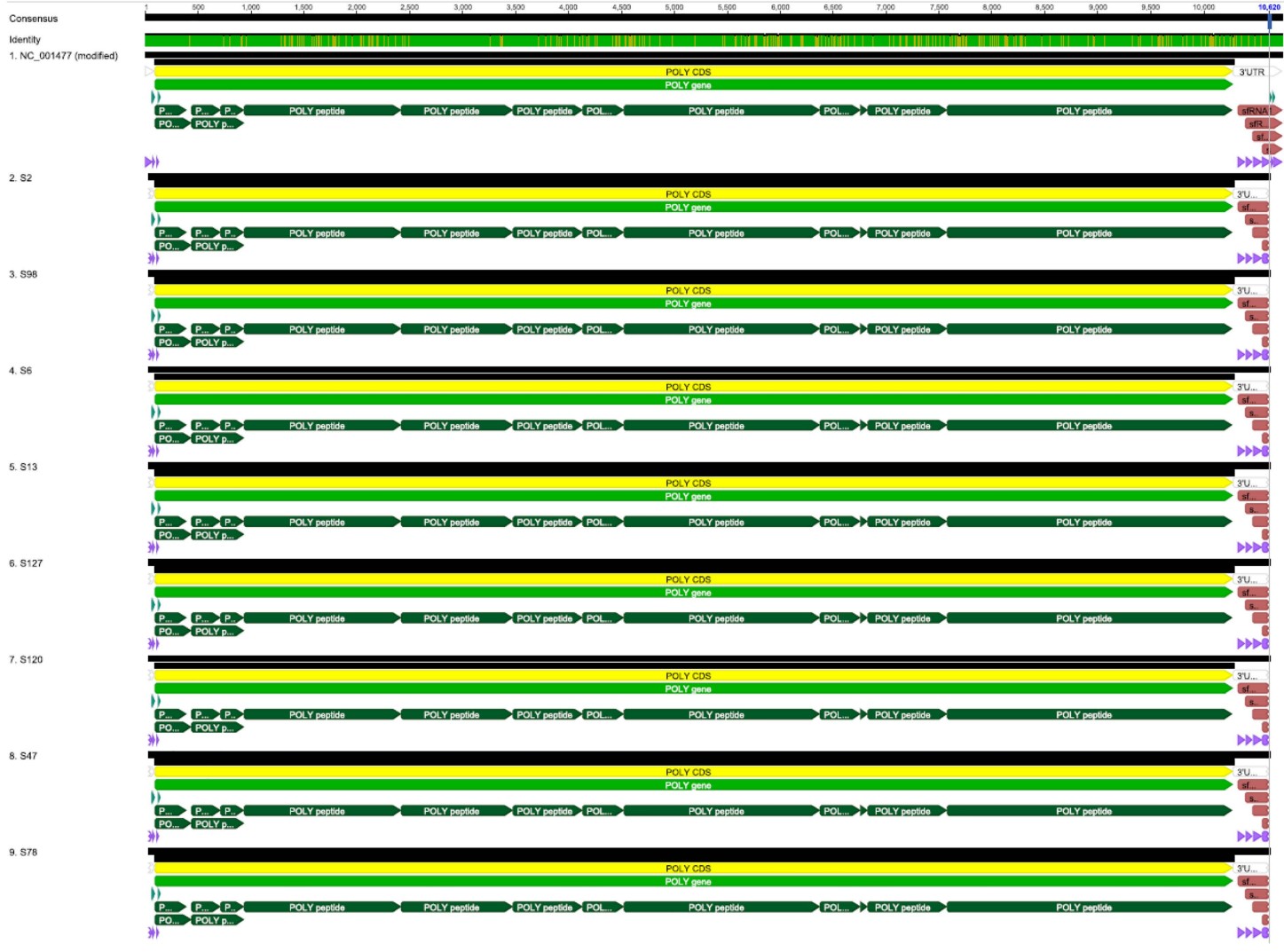

**Fig 5. Multiple sequence alignment of the assembled draft genomes with the reference DENVV-1 genome (NC_001477).**

and death. Since dengue has no proven treatment, supportive care is the only option [25]. Our study sought to elucidate factors associated with dengue illness and evaluate the roles of stress hormones as biomarkers for disease progression in relation to different DENV serotypes.

In our study we first determined serotypes of patients infected with dengue. Determining the DENV serotype is important for two key reasons: (i) more pathogenic DENV serotypes have been connected to risk factors for severe dengue; and (ii) the presence of several DENV serotypes in primary and secondary infections is assumed to be a risk factor for severe disease [26]. Our serotyping analysis identified DENV-1 as the predominant strain in Karachi, infecting 80% of patients, with DENV-2 at 16% and DENV-3 at a rare 0.7%. No cases of DENV-4 were detected. Although our cohort was based in Karachi rather than Mexico, the report by Mendoza-Cano et al. (2025) highlights that dengue serotype distributions are dynamic and that serotypes such as DENV-3 are re-emerging in different regions [27]. Our study relied exclusively on blood samples collected at a tertiary care hospital and included only laboratory blood reports, without access to extended clinical data. Consequently, dengue infections that do not lead to hospital presentation are not biologically captured in our

dataset. This directly explains the very low representation of DENV-3 in our cohort (0.7%) 1 out of 150 dengue-positive cases) and precludes statistically meaningful analysis of serotype-specific physiological and stress-hormone responses. Thus, while emerging serotypes may be circulating in the population, the hospital-restricted sampling framework limits our ability to investigate host hormonal responses to these newly circulating strains. Our findings also indicate that men made up the majority of the afflicted group. This finding aligns with prior demographic data indicating that males are linked to an increased risk of contracting dengue [28,29]. This is linked to exposure variations based on gender, such as time spent away from home. One possible explanation for the decreased incidence observed in women in nations such as Pakistan might be statistical artefacts associated with lesser reporting and care seeking for women from conventional healers who fail to submit data to public surveillance systems.

Regardless of the gender impacted, treating a disease needs a deeper comprehension of it. Our goal was to explore patterns of stress hormone response in dengue infection and assess whether these hormones show potential as descriptive biomarkers in relation to infection severity or viral serotype. For this purpose, we selected two stress hormones, cortisol and epinephrine.

It is well established that serum cortisol can indicate the severity of a major illness. Cortisol has been shown to be a biomarker for several conditions, including sepsis-septic shock, cardiovascular disease, and mental illness [30–32]. Our research observed that cortisol levels were lower in dengue patients compared to healthy controls. This trend contrasts with findings in COVID-19 and severe dengue cases where cortisol levels are reportedly higher [33]. A study reveals that cortisol levels are significantly greater in severe dengue patients, but as the virus reaches the deference stage, the cortisol levels drop. One possible explanation could be the timing of sample collection in our cohort. Alternatively, molecular mimicry mechanisms may induce low adrenocorticotropic hormone (ACTH) production, leading to decreased cortisol output a hypothesis that warrants further exploration [11]. Importantly, no significant differences in cortisol levels were observed across the various DENV serotypes.

We also examined adrenaline, a stress hormone that has been scarcely studied in dengue infection. Our results showed lower epinephrine levels in dengue patients compared to controls, with relatively lower levels in those co-infected with DENV-1 and DENV-2. Previous studies in other viral systems, including enterovirus 71 and herpes simplex virus, have reported that catecholamines can modulate viral replication and infectivity [34,35]; however, these findings cannot be directly extrapolated to dengue virus. In our cohort, only a very weak correlation was observed between viral load and adrenaline levels, which does not support a definitive role for adrenaline in DENV replication and highlights the need for targeted mechanistic studies in dengue models.

Similarly, there was no significant difference in blood parameters of DENV serotypes. The lowest median platelet count (thrombocytopenia) which is the hallmark of dengue infection was found with DENV-2, this makes this serotype the most potent among other serotypes identified. Other research groups have also found DENV-2 responsible for low platelet [36,37]. Platelet count was high when co-infected with DENV-1and 2. Seneratne et al. reported similar findings and hypothesized that this may be caused by a phenomenon "super-infection exclusion" where homologous virus co-infections may mitigate disease severity [38].

In general, a higher viral load has been linked to both illness prediction and severity [39]. In our investigation, we discovered a correlation between the measured blood parameters and stress hormones and virus load. Hemoglobin levels showed statistically significant inverse relationship with viral load. The most likely explanation is that the dengue virus may be sequestered in other organs such as the liver, spleen, kidneys, which may impact plasma viremia levels. Conflicting results may also be caused by differences in the sample duration, size, and infecting DENV serotype [40]. Despite contradicting study findings, the idea that dengue viral load is a significant determinant of illness severity remains questionable.

Whole-genome sequencing and phylogenetic analysis indicate that DENV-1 in Pakistan results from multiple introductions rather than a single lineage, potentially informing therapy strategies from various regions. Unlike our previous study, which focused solely on genomic characterization and phylogenetic diversity of DENV-1 strains in Pakistan [14], this work

integrates quantification of stress hormones (cortisol and adrenaline) and hematological parameters, correlating them with serotype and viral load. This virologic-endocrine approach provides novel insights into host response and highlights the interplay between viral diversity and host hormonal dynamics, a framework not previously reported in Pakistan.

Our study has certain limitations. First, the control group size was relatively small (n = 15) compared to the patient group (n = 150). Controls were recruited from healthy individuals during the outbreak period, but resource constraints limited their number. As a result, the study may have been underpowered to detect smaller associations, although it provided adequate data for descriptive analyses. Therefore, the findings should be interpreted as exploratory. Future larger, multi-center studies with formal sample size calculations will be needed to confirm and extend these preliminary observations. Second, detailed phase-specific fever data were unavailable, which restricted our ability to evaluate fluctuations in stress hormone levels across different stages of illness. While demographics, hematological indices, study location, and study period were reported, unmeasured confounders such as timing of sample collection, comorbidities, and concurrent medications may have influenced cortisol and adrenaline levels. Furthermore, the study was conducted during a peak outbreak session attending the government regulated tertiary care hospital. Though this may limit the external validity and generalizability of our findings to other populations or epidemiological contexts, Dow University Hospital serves to cater multi-ethnic patients from wider localities. Although these findings cannot be generalized to a global population, they do not appear to be confined to any particular sector or subgroup. Taken together, these limitations underscore the exploratory nature of our work. Nevertheless, examining the interactions between stress hormones and dengue serotypes may provide insights that may contribute to body of exploratory biomarker research.

## Conclusion

Our findings align with DENV-1and DENV-2 being prevalent in the Pakistani population, with DENV-3 less common and DENV-4 undetected. In wrapping up our study, we observed that patients infected with DENV-2 had the lowest platelet counts, suggesting this serotype may be associated with greater hematological impact in our cohort. Co-infection with DENV-1and 2 led to an increase in platelet count. While modest correlations were observed between blood parameters, stress hormones, and viral load, the cross-sectional design limits conclusions regarding their impact on illness severity. Stress and virus interaction in Dengue infection is complex. Our study indicates a drop in cortisol levels in dengue patients, unlike COVID-19 survivors with higher levels. Furthermore, phylogenetic examination of DENV-1 reveals that the Pakistani population does not possess its own variants, but rather has variations from other regions of the world. Adrenaline's role in dengue remains underexplored, offering valuable insights into stress hormones during infection. These findings offer preliminary insights into the interplay of dengue serotypes and may serve as a basis for generating hypotheses about their implications for infection outcomes.

## Supporting information

**S1 Fig. Maximum likelihood tree showing phylogenetic relationships among DENVV-1 samples.**
(PDF)

**S1 File. Supporting Information 01.**
(XLSX)

**S2 File. Supporting Information 02 revised.**
(XLSX)

## Author contributions

**Conceptualization:** Tehseen Fatima, Salman Ahmed Khan.

**Data curation:** Misbah Anwar, Abdul Waheed.

**Formal analysis:** Sadaf Khan, Muhammad Aurongzeb.

**Funding acquisition:** Mohammed Saeed Quraishy.

**Investigation:** Abdul Waheed, Mariyam Shahzad.

**Methodology:** Abdul Waheed, Tehseen Fatima, Mariyam Shahzad, Talat Roome, Zohaib Ul Hassan.

**Project administration:** Sadaf Khan, Muhammad Aurongzeb, Mohammed Saeed Quraishy.

**Resources:** Talat Roome, Ibrar Ahmed, Seil Kim, Mohammed Saeed Quraishy.

**Software:** Misbah Anwar, Sadaf Khan, Seil Kim, Zohaib Ul Hassan.

**Supervision:** Tehseen Fatima, Salman Ahmed Khan.

**Validation:** Misbah Anwar, Sadaf Khan, Ibrar Ahmed, Seil Kim, Salman Ahmed Khan.

**Visualization:** Ibrar Ahmed.

**Writing – original draft:** Abdul Waheed, Sadaf Khan, Muhammad Aurongzeb, Salman Ahmed Khan.

**Writing – review & editing:** Tehseen Fatima, Salman Ahmed Khan.

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
