## [Decision Letter · Decision Letter 0]

13 Aug 2025

Dear Dr. Khan,

Thank you for submitting your manuscript to PLOS ONE. After careful consideration, we feel that it has merit but does not fully meet PLOS ONE’s publication criteria as it currently stands. Therefore, we invite you to submit a revised version of the manuscript that addresses the points raised during the review process.

We look forward to receiving your revised manuscript.

Kind regards,

Harapan Harapan, MD, PhD

Academic Editor

PLOS ONE

Journal Requirements:

https://journals.plos.org/plosone/s/file?id=ba62/PLOSOne_formatting_sample_title_authors_affiliations.pdf....

3. Please update your submission to use the PLOS LaTeX template. The template and more information on our requirements for LaTeX submissions can be found at http://journals.plos.org/plosone/s/latex....

Reviewers' comments:

Reviewer's Responses to Questions

**Comments to the Author**

1. Is the manuscript technically sound, and do the data support the conclusions?

Reviewer #1: Yes

Reviewer #2: Partly

Reviewer #3: Partly

2. Has the statistical analysis been performed appropriately and rigorously?

Reviewer #1: Yes

Reviewer #2: No

Reviewer #3: No

3. Have the authors made all data underlying the findings in their manuscript fully available?

Reviewer #1: Yes

Reviewer #2: No

Reviewer #3: Yes

4. Is the manuscript presented in an intelligible fashion and written in standard English?

Reviewer #1: Yes

Reviewer #2: Yes

Reviewer #3: Yes

Reviewer #1: it is dscussing an interesting novel topic (DENV Serotype-Genotype Profiling and Its Association with Primary Stress Hormone in Dengue-Positive Patients). the manuscript is well designed and well written. the introduction, methods, results and discussion are well written and well presentes with good flaw of ideas.

Reviewer #2: The manuscript titled DENV Serotype-Genotype Profiling and Its Association with Primary Stress Hormone in Dengue-Positive Patients" reports on results from an observational study that aiming to explore whether there is a relationship between DENV serotypes/genotypes and stress hormone levels (specifically cortisol and adrenaline) in dengue-positive patients in Karachi, Pakistan. I have some concerns with the manuscript in its current version, please refer the followings:

1. The title suggests a mechanistic or causal relationship ("association") but the study design does not support causality—this is a cross-sectional observational study. Please adjust the title to reflect the descriptive nature of the analysis, e.g., “Profiles of Stress Hormones in Relation to DENV Serotypes among Dengue-Positive Patients.”

2. The abstract promises a novel exploration of hormonal response to DENV genotypes, but the results do not show statistically meaningful differences. Furthermore, it claims “modest positive correlation” but does not substantiate it with statistical significance or effect sizes. Please revise the abstract to reflect the exploratory and non-significant nature of the findings, and include correlation coefficients with p-values or confidence intervals where possible.

3. Large portions of the epidemiological background in the Introduction are devoted to general dengue epidemiology, which adds little to the justification of the study. Please streamline the background and focus instead on highlighting gaps in the literature related to host hormonal responses by serotype or genotype.

4. The rationale behind linking stress hormones (cortisol, adrenaline) with specific DENV genotypes is not grounded in any referenced theoretical or biological framework. There is no citation suggesting that serotype-specific viral behavior would mechanistically affect the HPA axis. Thus, authors must provide a mechanistic hypothesis or at least cite prior work showing links between DENV variants and neuroendocrine responses to strengthen their premise.

5. Previous studies on hormonal response in infectious diseases exist, but this work neither builds on nor properly distinguishes itself from them. So what differentiates this study must be clearly articulated—whether it’s geographic specificity, inclusion of whole-genome phylogenetics, or a novel analytic approach.

6. This is a descriptive, cross-sectional design, which is not sufficient to make any inference on interaction or impact between hormones and genotypes. Thus, conclusions implying serotype-hormone effects should be reworded to emphasize association without assuming directionality or causality.

7. There's no sample size justification or power calculation, making it unclear whether the study is adequately powered to detect the hypothesized associations. Please include a power analysis to justify whether the study was sufficiently powered to detect differences in hormone levels between serotypes.

8. The study was done in a limited geographic region and over a short time period. External validity is not addressed. This could limit generalizability to other settings or populations. Please clarify this limitation in the discussion section.

9. There is an ambiguity in patient selection and controls. The description lacks clarity on how negative controls were defined and matched. Thus, authors must clarify inclusion/exclusion criteria for controls, matching strategy (if any), and whether potential confounders were balanced between groups.

10. The use of correlational analysis with vague effect sizes and no correction for multiple testing is scientifically weak. For example, reporting weak positive correlations without FDR or Bonferroni correction inflates type I error. Authors may fix this by applying correction methods and reporting adjusted p-values.

11. The discussion implies biological importance of observed hormone differences without theoretical or empirical support. This could mislead readers into overinterpreting minor variations. Please tone down speculative interpretations unless supported by literature or strong statistical evidence.

12. Please improve the comparison with prior literature. Particularly, the manuscript does not engage with earlier studies on hormonal changes in dengue, missing an opportunity to contextualize findings in light of immune-endocrine interactions in viral infections.

13. I think part of the discussion is speculative in interpretation. The phylogenetic inference about multiple introductions of DENV into Pakistan does not align with the main hypothesis and seems tangential. To address this, authors may either connect the phylogenetic insights to disease severity or patient outcomes or move this section to supplementary results if not central to the main research question.

14. Authors claim about potential application in outbreak management. Is this accurate? Given the study design is observational in nature, and most of the findings are hard to implicate to somewhat actionable, this claim should be reworded to suggest that findings are hypothesis-generating rather than directly informing public health strategies.

15. Why do you think, or not, the study can warrant changes in diagnostic, clinical, or public health approaches? The current data are not robust or generalizable enough to warrant such changes. However, it may contribute to the body of exploratory biomarker research pending validation in larger, multi-site cohorts.

16. I worry about the use of only non-parametric tests limits analytical depth, and there's no multivariate control for confounders. Authors should consider multivariable regression models adjusting for age, sex, infection status, and time since symptom onset to better elucidate relationships.

17. More details should be added to patient demographics potentially influencing cortisol/adrenaline levels (e.g., time of sample collection, medications, comorbidities). These factors could significantly confound hormone measurements and must be accounted for or discussed as limitations.

18. Consider to add more relevant studies in the discussion, such as by stating: "Although our cohort is based in Karachi rather than Mexico, global shifts in serotype prevalence—as documented by Mendoza‑Cano et al. (2025)—underscore the relevance of investigating whether new serotypes (e.g., DENV‑3) elicit different host physiological responses.” Ref: PLOS One. 2025 May 22;20(5)\:e0324754

19. Introduction can benefit from the addition of: “As example in Indonesia, dengue control efforts are increasingly compromised by widespread resistance of Aedes aegypti to standard insecticides, notably pyrethroids and organophosphates—especially across Java and neglected rural areas—highlighting the need for sustainable alternatives such as plant-based biolarvicides (Kasman et al., 2025) \[e‑1819]”

20. To lay the biological foundation for exploring hormonal vs. immune biomarker patterns, authors may incorporate the following: “Elevated pro-inflammatory cytokines such as TNF‑α, IL‑6, and IL‑17, along with differential IL‑10 responses in dengue hemorrhagic versus classical dengue, highlight the immune axis in severity progression—supporting theoretical rationale for exploring stress hormone–cytokine interplay (Masyeni et al., Narra J 2024;4(1)\:e309)”

21. Authors have not made the underlying data publicly available as recommended by PLOS ONE. This contradicts the journal’s data availability policy, which emphasizes transparency and reproducibility. Authors must deposit the complete dataset in a public repository or provide access upon request, along with metadata and a clear data availability statement in the manuscript.

Reviewer #3: - Address Unequal Group Sizes:

The unbalanced distribution of serotypes (DENV-1: 80%; DENV-2: 16%; DENV-3: 0.7%; co-infections: 3.3%) may limit statistical power for comparisons involving DENV-3 and co-infections. Implement post-hoc tests (e.g., Dunn’s test) following the Kruskal-Wallis test to identify specific differences between serotypes, accounting for unequal group sizes.

- Adjust for Multiple Comparisons:

The study conducts multiple comparisons (e.g., hematological parameters across serotypes). To reduce the risk of Type I errors, apply corrections such as the Bonferroni or False Discovery Rate (FDR) method for multiple tests, ensuring more reliable p-values.

- Increase Control Group Size:

The control group (n=15) is small compared to the patient group (n=150), potentially limiting the power of comparisons (e.g., Mann-Whitney U tests). Increasing the control group size would improve the robustness of differences observed in stress hormone levels

- Incorporate Phase-Specific Analysis:

The absence of fever phase data is a noted limitation. Stratify analyses by disease phase (febrile, critical, or recovery) to explore variations in stress hormone levels, which could strengthen the interpretation of cortisol and epinephrine findings.

- Report Phylogenetic Confidence Measures:

The phylogenetic analysis lacks details on node confidence (e.g., bootstrap values). Include bootstrap or posterior probability values for the maximum likelihood tree generated by W-IQ-TREE to validate conclusions about multiple DENV-1 introductions.

- Discuss Clinical Relevance of Weak Correlations:

The Spearman correlations showed weak coefficients (e.g., r=0.184 for cortisol, r=-0.228 for hemoglobin). Provide a deeper discussion on the clinical significance of these weak associations and assess whether the sample size was sufficient to detect stronger relationships.

.

Reviewer #1: No

Reviewer #2: No

Reviewer #3: **Yes:**Samir Mansour Moraes CassebSamir Mansour Moraes CassebSamir Mansour Moraes CassebSamir Mansour Moraes Casseb

---

## [Author Response · Author response to Decision Letter 1]

19 Sep 2025

Response to Reviewers:

We would like to sincerely thank the reviewers for their time, effort, and valuable comments in reviewing our work. Their constructive feedback has greatly helped us refine and improve the manuscript, making it more meaningful and useful for the scientific community. Please find below our detailed responses to the reviewers’ comments, addressed point by point.

Reviewer No 02:

1.The title suggests a mechanistic or causal relationship ("association") but the study design does not support causality - this is a cross-sectional observational study. Please adjust the title to reflect the descriptive nature of the analysis, e.g., "Profiles of Stress Hormones in Relation to DENV Serotypes among Dengue-Positive Patients."

Response: Title, Page no. 2, Line 02

In accordance with the reviewer’s suggestion, we have revised the title to “Profiles of Stress Hormones in Relation to DENV Serotypes among Dengue-Positive Patients.”

2.The abstract promises a novel exploration of hormonal response to DENV genotypes, but the results do not show statistically meaningful differences. Furthermore, it claims "modest positive correlation" but does not substantiate it with statistical significance or effect sizes. Please revise the abstract to reflect the exploratory and non-significant nature of the findings, and include correlation coefficients with p-values or confidence intervals where possible.

Response: Abstract, Page no. 2, Line no. 47-52 & 56-58

We have revised the abstract to better reflect the exploratory and descriptive nature of the study. Specifically, we have now included correlation coefficients with p-values to substantiate the reported associations. “Notably, comparisons across different DENV serotypes did not reveal statistically significant differences in stress hormone response, indicating only descriptive trends. Spearman’s correlation analysis showed weak, non-significant associations between viral load and cortisol (r = 0.184, p = 0.170) and between viral load and epinephrine (r = 0.165, p = 0.224). Statistically significant but modest negative correlation was observed only between viral load and hemoglobin (r = -0.228, p = 0.005).” “However, as most associations were exploratory and not statistically significant. The descriptive trends identified here highlight the need for larger, longitudinal studies to better define the relationship between stress hormones, viral load, and dengue severity.” These values have been incorporated into the revised abstract and respective results section to provide full transparency regarding the non-significant nature of most associations.

3.Large portions of the epidemiological background in the Introduction are devoted to general dengue epidemiology, which adds little to the justification of the study. Please streamline the background and focus instead on highlighting gaps in the literature related to host hormonal responses by serotype or genotype.

Response: Introduction Section, Page no. 04, Line 104-116

In the revised manuscript, we have streamlined the epidemiological background to a concise summary and shifted the focus to highlighting gaps in the literature regarding host hormonal responses. “Although dengue has been widely studied in terms of epidemiology and viral diversity, little attention has been given to host endocrine responses. For example, cortisol has been linked to disease severity, with lower levels reported in children with severe dengue shock syndrome [10, 14], but such studies did not assess differences by serotype or genotype. To date, no research has systematically evaluated stress hormone responses (cortisol and adrenaline) across DENV serotypes. Addressing this gap, our study investigates stress hormone profiles in relation to DENV serotype-genotype characterization in dengue-positive patients from Pakistan.

We have previously reported the whole-genome phylogenetic characterization of DENV-1 strains circulating in Pakistan [14]. Building on that foundation, the present study takes a more integrative approach. Specifically, we investigate host endocrine responses (cortisol and adrenaline) in dengue-positive patients and compare these with viral load, serotype distribution, and hematological indices.”

To our knowledge, no study has systematically evaluated stress hormone dynamics (cortisol and adrenaline) across different DENV serotypes/genotypes. We have now emphasized this gap in the revised Introduction to better justify the rationale of our study.

4.The rationale behind linking stress hormones (cortisol, adrenaline) with specific DENV genotypes is not grounded in any referenced theoretical or biological framework. There is no citation suggesting that serotype-specific viral behavior would mechanistically affect the HPA axis. Thus, authors must provide a mechanistic hypothesis or at least cite prior work showing links between DENV variants and neuroendocrine responses to strengthen their premise.

Response: Introduction Section, Page no. 3-4, Line 93-100

We have now added new references to the Introduction to strengthen the scientific premise. “This interaction between viral infection, cytokine signaling, and HPA axis activation is well documented [9]. In dengue specifically, altered cortisol levels have been reported in relation to disease severity [10, 11], although prior studies have not examined variation across serotypes or genotypes. Elevated pro-inflammatory cytokines such as TNF-α, IL-6, and IL-17, together with differential IL-10 responses in dengue hemorrhagic versus classical dengue, have been shown to play a major role in severity progression. These findings underscore the immune axis in dengue pathogenesis and provide theoretical support for exploring potential interactions between stress hormones and cytokine responses [12].”

5.Previous studies on hormonal response in infectious diseases exist, but this work neither builds on nor properly distinguishes itself from them. So what differentiates this study must be clearly articulated whether it is geographic specificity, inclusion of whole-genome phylogenetics, or a novel analytic approach.

Response: Introduction Section, Page no. 04, line 104-116

We have revised the manuscript to explicitly highlight how the current study differs from both earlier work on hormonal responses in infectious diseases and from our own previously published genomic study. “Although dengue has been widely studied in terms of epidemiology and viral diversity, little attention has been given to host endocrine responses. For example, cortisol has been linked to disease severity, with lower levels reported in children with severe dengue shock syndrome [10-14], but such studies did not assess differences by serotype or genotype. To date, no research has systematically evaluated stress hormone responses (cortisol and adrenaline) across DENV serotypes. Addressing this gap, our study investigates stress hormone profiles in relation to DENV serotype-genotype characterization in dengue-positive patients from Pakistan.

We have previously reported the whole-genome phylogenetic characterization of DENV-1 strains circulating in Pakistan [14]. Building on that foundation, the present study takes a more integrative approach. Specifically, we investigate host endocrine responses (cortisol and adrenaline) in dengue-positive patients and compare these with viral load, serotype distribution, and hematological indices.”

(Discussion Section, Page no. 11, line 355-360)

“Unlike our previous study, which focused solely on genomic characterization and phylogenetic diversity of DENV-1 strains in Pakistan [14], this work integrates quantification of stress hormones (cortisol and adrenaline) and hematological parameters, correlating them with serotype and viral load. This virologic-endocrine approach provides novel insights into host response and highlights the interplay between viral diversity and host hormonal dynamics, a framework not previously reported in Pakistan.”

6.This is a descriptive, cross-sectional design, which is not sufficient to make any inference on interaction or impact between hormones and genotypes. Thus, conclusions implying serotype-hormone effects should be reworded to emphasize association without assuming directionality or causality.

Response: Results, discussion and conclusion sections

We have carefully revised the manuscript to avoid wording that could imply causality or directionality, in the following sections.

Results (line no. 266 - 267)

“Only the associations between viral load, and hemoglobin was found statistically significant (p < 0.05).”

Discussion (line no. 345 - 348)

“In our investigation, we discovered a correlation between the measured blood parameters and stress hormones and virus load. Hemoglobin levels showed a modest but statistically significant inverse relationship with viral load.”

Conclusion sections (line no. 382-387)

“In wrapping up our study, we observed that patients infected with DENV-2 had the lowest platelet counts, suggesting this serotype may be associated with greater hematological impact in our cohort.

While modest correlations were observed between blood parameters, stress hormones, and viral load, the cross-sectional design limits conclusions regarding their impact on illness severity.”

7There's no sample size justification or power calculation, making it unclear whether the study is adequately powered to detect the hypothesized associations. Please include a power analysis to justify whether the study was sufficiently powered to detect differences in hormone levels between serotypes.

Response: Discussion section, Page no. 12, line 361 - 367

A formal power calculation was not performed at the study’s inception, as this was an exploratory study conducted during a dengue outbreak. Instead, the study was designed to include all dengue-positive patients presenting to Dow University Hospital within the defined study period (September–December 2022), resulting in a total of 150 cases and 15 healthy controls. We recognize that the absence of a prior sample size justification limits the ability to claim adequate statistical power for detecting associations between stress hormone levels and DENV Serotypes.

To address this, we added the following:

“Our study has certain limitations. First, the control group size was relatively small (n = 15) compared to the patient group (n = 150). Controls were recruited from healthy individuals during the outbreak period, but resource constraints limited their number. As a result, the study may have been underpowered to detect smaller associations, although it provided adequate data for descriptive analyses. Therefore, the findings should be interpreted as exploratory. Future larger, multi-center studies with formal sample size calculations will be needed to confirm and extend these preliminary observations.”

8.The study was done in a limited geographic region and over a short time period. External validity is not addressed. This could limit generalizability to other settings or populations. Please clarify this limitation in the discussion section.

Response: Discussion Section, Page no. 12, Line no. 371 - 376

“Furthermore, the study was conducted during a peak outbreak session attending the government regulated tertiary care hospital. Though this may limit the external validity and generalizability of our findings to other populations or epidemiological contexts, Dow University Hospital serves to cater multi-ethnic patients from wider localities. Although these findings cannot be generalized to a global population, they do not appear to be confined to any particular sector or subgroup.”

9.There is an ambiguity in patient selection and controls. The description lacks clarity on how negative controls were defined and matched. Thus, authors must clarify inclusion/exclusion criteria for controls, matching strategy (if any), and whether potential confounders were balanced between groups.

Response: Materials and Methods, Page no. 5, line 129 - 135

We recognize the limitations and have clarified the selection criteria for cases and controls in the revised manuscript as “During the outbreak, dengue-positive cases were recruited following clinical suspicion by physicians and confirmed through NS1 antigen and RT-PCR testing. Negative controls were recruited from individuals testing negative for dengue in the same setting, using random selection according to resource availability. Detailed fever history was not systematically available, which we recognize as a limitation. Although one-to-one matching was not possible, group-level comparability in age and sex distribution was maintained, and baseline hematological indices were reviewed to minimize imbalances”.

10The use of correlational analysis with vague effect sizes and no correction for multiple testing is scientifically weak. For example, reporting weak positive correlations without FDR or Bonferroni correction inflates type I error. Authors may fix this by applying correction methods and reporting adjusted p-values.

Materials and Methods, Page no. 7, line no. 223-227

Table no 6, Page no, 27

Figure 3, Page no, 18

Figure 4 Page no, 19

Response:

As suggested, Bonferroni correction for multiple comparisons was applied to all correlational analyses, and Bonferroni-adjusted p-values are now reported. This has been added to the Statistical Analysis section, and the relevant table (Table 6) has also been updated. p<0.05 was considered statistically significant in all analysis such as Figure 3 & 4; Table 3 – 6.

11.The discussion implies biological importance of observed hormone differences without theoretical or empirical support. This could mislead readers into overinterpreting minor variations. Please tone down speculative interpretations unless supported by literature or strong statistical evidence.

Response:

As noted in our response to Comment 06, we have carefully revised the Results, Discussion, and Conclusion sections to remove causal or mechanistic language and to emphasize that our findings are exploratory associations rather than definitive biological effects. In line with this revision, speculative interpretations of hormone differences have been toned down, and the text now clearly frames these observations as preliminary and hypothesis-generating, requiring further validation in larger studies.

12.Please improve the comparison with prior literature. Particularly, the manuscript does not engage with earlier studies on hormonal changes in dengue, missing an opportunity to contextualize findings in light of immune-endocrine interactions in viral infections.

Response:

As noted in our response to Comment 04, we have revised the Introduction to incorporate references on immune–endocrine interactions in viral infections and earlier studies reporting cortisol dynamics in dengue patients. In addition, we have expanded the Discussion to compare our findings with these prior reports, while carefully framing our observations as exploratory associations (as also addressed under Comment 06 and Comment 11). Together, these revisions strengthen the contextualization of our study and clarify how our work contributes to the existing body of knowledge by addressing a gap not previously explored the association of stress hormone profiles with DENV serotypes/genotypes.

13. I think part of the discussion is speculative in interpretation. The phylogenetic inference about multiple introductions of DENV into Pakistan does not align with the main hypothesis and seems tangential. To address this, authors may either connect the phylogenetic insights to disease severity or patient outcomes or move this section to supplementary results if not central to the main research question.

Response: Page no 21

The phylogenetic inference, while informative, is not central to our primary hypothesis regarding stress hormone profiles. In response, we have moved the phylogenetic tree to the Supplementary Materials and streamlined the related discussion to a brief contextual note rather than a central interpretation. This ensures that the main text remains focused on the study’s core objectives, while still allowing readers access to the phylogenetic analysis for completeness. Supplementary Figur

---

## [Decision Letter · Decision Letter 1]

29 Dec 2025

Dear Dr. Khan,

Thank you for submitting your manuscript to PLOS ONE. After careful consideration, we feel that it has merit but does not fully meet PLOS ONE’s publication criteria as it currently stands. Therefore, we invite you to submit a revised version of the manuscript that addresses the points raised during the review process.

We look forward to receiving your revised manuscript.

Kind regards,

Harapan Harapan, MD, PhD

Academic Editor

PLOS One

**Journal Requirements:**

Reviewers' comments:

Reviewer's Responses to Questions

**Comments to the Author**

Reviewer #2: All comments have been addressed

Reviewer #4: (No Response)

2. Is the manuscript technically sound, and do the data support the conclusions?

Reviewer #2: Yes

Reviewer #4: No

3. Has the statistical analysis been performed appropriately and rigorously?

Reviewer #2: Yes

Reviewer #4: Yes

4. Have the authors made all data underlying the findings in their manuscript fully available?

Reviewer #2: Yes

Reviewer #4: Yes

5. Is the manuscript presented in an intelligible fashion and written in standard English?

Reviewer #2: Yes

Reviewer #4: Yes

**Reviewer #2:**Authors have addressed my previous comments, especially by avoiding the inferrence-like statement and explain the statistic power. This manuscript is acceptable for publication.Authors have addressed my previous comments, especially by avoiding the inferrence-like statement and explain the statistic power. This manuscript is acceptable for publication.Authors have addressed my previous comments, especially by avoiding the inferrence-like statement and explain the statistic power. This manuscript is acceptable for publication.Authors have addressed my previous comments, especially by avoiding the inferrence-like statement and explain the statistic power. This manuscript is acceptable for publication.

**Reviewer #4:**General concern:General concern:General concern:General concern:

The biggest limitation of this study is the lack of data on fever days. Dengue is a very dynamic disease, changing day by day and even hour by hour. Therefore, laboratory data is also very dynamic. For example, during the first 3 days of fever (the febrile phase), we do not expect thrombocytopenia, and leukopenia usually occurs. Hemoglobin levels are also indirectly related; what is more accurately observed is the dynamics of hematocrit. There is also no data related to disease severity (with warning signs or severe dengue, although my assumption is that if hospitalized, they are not patients without warning signs). Stress hormones are certainly highly related to the dynamics of dengue disease progression and its severity. This is also what might cause the results to not show significance. Although each serotype may have varying severity, it also depends on the course of the illness (course of day-fever).

Questions:

In what way exactly is there a direct logical connection from discussing the newly emerging serotypes (such as DENV-3), as cited in Mendoza-Cano et al. in 2025, to the following assertion about self-managed and unreported cases of dengue infection occurring in Pakistan? And precisely how does this issue of underreporting thwart research efforts into identifying host physiological and hormonal responses to newly emerging serotypes in Pakistan?

2. After being presented as a hypothesis, were there any particular clinical findings, characteristics of the virus in your population, or other pilot data to which you wished to draw attention as to why molecular mimicry would be considered as an alternative explanation for the lower cortisol levels?

3. In regards to it being observed that your sources encompass research involving HSV and E71 but not necessarily related to DENV, what exactly would be the biologic rationale or evidential source from literature supporting such a postulate particular to the function of adrenaline in DENV replication? In addenda to this question, what would be the strength of your assertion about "adrenaline not contributing" to DENV replication based exclusively on "very weak correlation" to viral load?

4. Do patients co-infected with both DENV-1 and DENV-2 who had greater numbers of platelets have data related to other clinical features (such as severity grade, rate of hospitalizations due to severe dengue infection, ICU admissions rate, time to recovery) collected to provide direct supportive data to your hypothesis about "super-infection exclusion" contributing to reduced disease severity?

5. Although it would be difficult to generalize to other regions internationally, what specific data suggests "the results do not appear to be confined to any particular sector or subgroup" in Karachi despite being carried out in a single tertiary-level hospital environment in the midst of a major outbreak?

.

Reviewer #2: No

Reviewer #4: No

---

## [Author Response · Author response to Decision Letter 2]

21 Jan 2026

Response Sheet

We would like to sincerely thank the reviewers and editor for their time, effort, and valuable comments in reviewing our work. Their constructive feedback has greatly helped us refine and improve the manuscript, making it more meaningful and useful for the scientific community. Please find below our detailed responses to the reviewers’ comments addressed point by point.

Reviewer No 04:

General concern from the reviewer: The biggest limitation of this study is the lack of data on fever days. Dengue is a very dynamic disease, changing day by day and even hour by hour. Therefore, laboratory data is also very dynamic. For example, during the first 3 days of fever (the febrile phase), we do not expect thrombocytopenia, and leukopenia usually occurs. Hemoglobin levels are also indirectly related; what is more accurately observed is the dynamics of hematocrit. There is also no data related to disease severity (with warning signs or severe dengue, although my assumption is that if hospitalized, they are not patients without warning signs). Stress hormones are certainly highly related to the dynamics of dengue disease progression and its severity. This is also what might cause the results to not show significance. Although each serotype may have varying severity, it also depends on the course of the illness (course of day-fever).

Author’s Response: Availability of extended clinical profile was a major limitation of the study. Since the study was conducted in a tertiary care hospital and the patients were recruited from the outpatient’s clinic, a complete record of clinical profile including fever and hospitalizations could not be managed. And for similar reasons it could potentially result in the variability of the laboratory data which cannot be mapped with the fever profile.

We agree that the relevance of stress hormones is not significant and that would potentially be due to the variation in available confounding factors including stages of the disease (early onset v/s recovery stage). However, the study would serve as the generation of the pilot data and would potentially serve as refinement of the second stage of the project.

Questions:

Reviewer’s Question Q1. In what way exactly is there a direct logical connection from discussing the newly emerging serotypes (such as DENV-3), as cited in Mendoza-Cano et al. in 2025, to the following assertion about self-managed and unreported cases of dengue infection occurring in Pakistan? And precisely how does this issue of underreporting thwart research efforts into identifying host physiological and hormonal responses to newly emerging serotypes in Pakistan?

Author’s Response to Q1: Although our cohort was based in Karachi, the report by Mendoza-Cano et al. (2025) highlights that dengue serotype distributions are dynamic and that serotypes such as DENV-3 are re-emerging in different regions. Our study relied exclusively on blood samples collected at a local tertiary care hospital and included only laboratory blood reports, without access to extended clinical or community-level data. Consequently, dengue infections that do not lead to hospital presentation are not biologically captured in our dataset. This directly explains the very low representation of DENV-3 in our cohort (1 out of 150 dengue-positive cases), and therefore no serotype-specific hormonal correlation analysis was performed for DENV-3. Thus, although emerging serotypes may be circulating in the population, the hospital-restricted sampling framework limits the ability to investigate host physiological and stress-hormone responses to these newly circulating strains.

The following information has been added to the discussion section page number Page no.10, Line 314-324

“Although our cohort was based in Karachi rather than Mexico, the report by Mendoza-Cano et al. (2025) highlights that dengue serotype distributions are dynamic and that serotypes such as DENV-3 are re-emerging in different regions [27]. Our study relied exclusively on blood samples collected at a tertiary care hospital and included only laboratory blood reports, without access to extended clinical data. Consequently, dengue infections that do not lead to hospital presentation are not biologically captured in our dataset. This directly explains the very low representation of DENV-3 in our cohort (0.7%) i.e. 1 out of 150 dengue-positive cases) and precludes statistically meaningful analysis of serotype-specific physiological and stress-hormone responses. Thus, while emerging serotypes may be circulating in the population, the hospital-restricted sampling framework limits our ability to investigate host hormonal responses to these newly circulating strains.”

Reviewer’s Question 2.

Author’s Response to Q2. The reported study does not revolve around the characterization of the pathogen else than the molecular profile in terms of serotype and genotype. Although it would be convincing to relate the molecular profile of the virus to the cortisol level, a direct inference is not possible as the cortisol level can be modulated by the range of factors including patient own stress profile.

Reviewer’s Question Q3.

In regard to it being observed that your sources encompass research involving HSV and E71 but not necessarily related to DENV, what exactly would be the biologic rationale or evidential source from literature supporting such a postulate particular to the function of adrenaline in DENV replication? In addenda to this question, what would be the strength of your assertion about "adrenaline not contributing" to DENV replication based exclusively on "very weak correlation" to viral load?

Author’s Response to Q3.

We agree that studies on HSV and enterovirus 71 cannot be directly extrapolated to DENV, as viral tropism, replication compartments, and host–virus interactions differ substantially. These studies were cited only to highlight that catecholamines can modulate viral replication in some viral systems, not as evidence of a similar mechanism in dengue.

Furthermore, we acknowledge that the non-significant correlation observed between viral load and adrenaline levels in our cohort does not justify a definitive conclusion that adrenaline does not contribute to DENV replication. We have therefore revised the text to remove this assertion and now state that our data do not provide evidence supporting a role for adrenaline in DENV replication, and that mechanistic studies are required to address this question.

The following information has been added to the discussion section page number Page no. 11, Line 346-354

“We also examined adrenaline, a stress hormone that has been scarcely studied in dengue infection. Our results showed lower epinephrine levels in dengue patients compared to controls, with relatively lower levels in those co-infected with DENV-1 and DENV-2. Previous studies in other viral systems, including enterovirus 71 and herpes simplex virus, have reported that catecholamines can modulate viral replication and infectivity [34, 35]; however, these findings cannot be directly extrapolated to dengue virus. In our cohort, only a very weak correlation was observed between viral load and adrenaline levels, which does not support a definitive role for adrenaline in DENV replication and highlights the need for targeted mechanistic studies in dengue models.”

Reviewer’s Question Q4.

Do patients co-infected with both DENV-1 and DENV-2 who had greater numbers of platelets have data related to other clinical features (such as severity grade, rate of hospitalizations due to severe dengue infection, ICU admissions rate, time to recovery) collected to provide direct supportive data to your hypothesis about "super-infection exclusion" contributing to reduced disease severity?

Author’s Response to Q4.

As mentioned above, the samples were collected from patients attending walk-in clinics, details relevant to ICU admissions or hospitalization could not be related.

Reviewer’s Question Q5.

Although it would be difficult to generalize to other regions internationally, what specific data suggests "the results do not appear to be confined to any particular sector or subgroup" in Karachi despite being carried out in a single tertiary-level hospital environment in the midst of a major outbreak?

Author’s Response to Q5.

Although the samples have been collected from a tertiary care hospital in Karachi, the data can be generalized as Karachi is one of the largest metropolitan sector of the country and caters a mass population of different background, multiple ethnicities and socioeconomic groups.

---

## [Decision Letter · Decision Letter 2]

22 Feb 2026

Profiles of Stress Hormones in Relation to DENV Serotypes among Dengue-Positive Patients

PONE-D-25-27769R2

Dear Dr. Khan,

We’re pleased to inform you that your manuscript has been judged scientifically suitable for publication and will be formally accepted for publication once it meets all outstanding technical requirements.

Kind regards,

Harapan Harapan, MD, PhD

Academic Editor

PLOS One

Additional Editor Comments (optional):

Reviewers' comments:

Reviewer's Responses to Questions

**Comments to the Author**

Reviewer #2: All comments have been addressed

Reviewer #4: All comments have been addressed

2. Is the manuscript technically sound, and do the data support the conclusions?

Reviewer #2: Yes

Reviewer #4: (No Response)

3. Has the statistical analysis been performed appropriately and rigorously?

Reviewer #2: Yes

Reviewer #4: (No Response)

4. Have the authors made all data underlying the findings in their manuscript fully available?

Reviewer #2: Yes

Reviewer #4: (No Response)

5. Is the manuscript presented in an intelligible fashion and written in standard English?

Reviewer #2: Yes

Reviewer #4: (No Response)

Reviewer #2: The authors have thoroughly addressed my previous comments. They have appropriately revised the manuscript to avoid inferential overstatement and have clearly discussed its limitations, including the statistical power. The manuscript is now acceptable for publication.

Reviewer #4: (No Response)

.

Reviewer #2: No

Reviewer #4: No

---

## [Editor Report · Acceptance letter]

PONE-D-25-27769R2

PLOS One

Dear Dr. Khan,

I'm pleased to inform you that your manuscript has been deemed suitable for publication in PLOS One. Congratulations! Your manuscript is now being handed over to our production team.

Kind regards,

on behalf of

Dr. Harapan Harapan

Academic Editor

PLOS One